# Do the Differences in the Epiligament of the Proximal and Distal Parts of the Anterior Cruciate Ligament Explain Their Different Healing Capacities? Quantitative and Immunohistochemical Analysis of CD34 and α-SMA Expression in Relation to the Epiligament Theory

**DOI:** 10.3390/biomedicines12010156

**Published:** 2024-01-11

**Authors:** Georgi P. Georgiev, Yordan Yordanov, Łukasz Olewnik, Richard Shane Tubbs, Robert F. LaPrade, Julian Ananiev, Svetoslav A. Slavchev, Iva N. Dimitrova, Lyubomir Gaydarski, Boycho Landzhov

**Affiliations:** 1Department of Orthopedics and Traumatology, University Hospital Queen Giovanna-ISUL, Medical University of Sofia, 1527 Sofia, Bulgaria; 2Department of Pharmacology, Pharmacotherapy and Toxicology, Faculty of Pharmacy, Medical University of Sofia, 1000 Sofia, Bulgaria; yyordanov@pharmfac.mu-sofia.bg; 3Department of Anatomical Dissection and Donation, Chair of Anatomy and Histology, Medical University of Lodz, 90-419 Lodz, Poland; lukasz.olewnik@umed.lodz.pl; 4Department of Anatomical Sciences, St. George’s University, St. George 1473, Grenada; shane.tubbs@icloud.com; 5Department of Neurosurgery, Tulane University School of Medicine, New Orleans, LA 70112, USA; 6Department of Neurology, Tulane University School of Medicine, New Orleans, LA 70112, USA; 7Department of Structural and Cellular Biology, Tulane University School of Medicine, New Orleans, LA 70112, USA; 8Department of Surgery, Tulane University School of Medicine, New Orleans, LA 70112, USA; 9Twin Cities Orthopedics, Edina, MN 55435, USA; laprademdphd@gmail.com; 10Department of General and Clinical Pathology, Faculty of Medicine, Trakia University, 6000 Stara Zagora, Bulgaria; operation@abv.bg; 11University Hospital of Orthopedics “Prof. B. Boychev”, Medical University of Sofia, 1614 Sofia, Bulgaria; s.slavchev@medfac.mu-sofia.bg; 12Department of Cardiology, University Hospital “St. Ekaterina”, Medical University of Sofia, 1431 Sofia, Bulgaria; dimytrova@yahoo.com; 13Department of Anatomy, Histology and Embryology, Medical University of Sofia, 1431 Sofia, Bulgaria; lgaidarsky@gmail.com (L.G.); landzhov_medac@abv.bg (B.L.)

**Keywords:** epiligament, epiligament theory, anterior cruciate ligament, medial collateral ligament, knee, human

## Abstract

The aim of this study was to assess the epiligament theory by determining the normal epiligament morphology of the proximal and distal parts of the anterior cruciate ligament in humans and analyzing the differences between them and the midportion of the ligament in terms of cell numbers and expression of CD34 and α-SMA. Samples were obtained from the anterior cruciate ligaments of 12 fresh knee joints. Monoclonal antibodies against CD34 and α-SMA were used for immunohistochemistry. Photomicrographs were analyzed using ImageJ software, version 1.53f. The cell density was higher in the epiligament than in the ligament connective tissue. Cell counts were higher in the proximal and distal thirds than in the midsubstance of the epiligament. CD34 was expressed similarly in the proximal and distal thirds, although it seemed slightly more pronounced in the distal third. α-SMA expression was more robust in the proximal than the distal part. The results revealed that CD34 and α-SMA are expressed in the human epiligament. The differences between the numbers of cells in the proximal and distal parts of the epiligament and the expression of CD34 and α-SMA enhance epiligament theory. Future investigations into improving the quality of ligament healing should not overlook the epiligament theory.

## 1. Introduction

Lesions of the anterior cruciate ligament (ACL) are common [1]. ACL injuries reflect recent and more long-term activities; they are considered risk factors for future osteoarthritic changes in the knee [2]. Formerly, primary repair of the ACL injury was performed, but no healing was established in more than 90% of cases [2,3]. Subsequently, ACL reconstruction with a tendon graft was accepted as the gold standard of treatment, especially for active patients who wanted to return to sports [3,4]. Nevertheless, progressive articular cartilage degeneration was established [3,5]. Therefore, different studies have suggested alternative methods, such as repair and regeneration of the ACL [3,4]. This renewed interest in repairing could have advantages such as preserved proprioception, less invasiveness, no need for grafting, faster recovery of the range of motion, and decreased knee awareness [4]. Repair is indicated for patients with a proximal injury of the ACL [4]. Several recent cohort studies have given encouraging results for repair techniques [6,7,8,9]. Tibial-sided soft-tissue avulsion of the ACL has also been reported, though it is rare [1]. The treatment could be reinsertion of the ACL using a transosseous pull-out repair. Drilling of tibial tunnels to pull out the loop-stitches, in effect, delivers stem cells and growth factors to the site of injury and could thereby stimulate the healing process [1].

A recently introduced theory, known as the epiligament (EL) theory, is grounded in studies that delineate the EL (epi-[Greek—on or upon]; ligament [Latin—ligare, to bind]) as a thin layer of connective tissue adherent to the ligament [10,11]. In essence, the EL theory explains the lack of healing observed in injuries in the middle-third of the ACL. According to this theory, the EL serves as the primary source of cells, vessels, and molecules essential for the healing process after ligament injuries, as proposed by Georgiev et al. [10]. Expanding on this theory, Georgiev et al. conducted a comparative analysis of EL tissues in the MCL and ACL [11]. While noting similar morphological characteristics of the EL in both ligaments, they identified a significant difference in cell quantity, with the MCL exhibiting a higher cell count, potentially explaining its superior healing capacity [11]. Exploring the morphological features of the EL and recognizing significant differences in cell and molecule numbers responsible for healing (including major collagen types, matrix metalloproteinases, fibronectin, CD34, VEGF, α-SMA), the EL theory was further refined, providing insights into the challenges of ACL healing in the midsubstance [12,13,14].

The blood vessel walls are acknowledged as a plentiful source of stem cells expressing CD34 [15,16]. A recent study by Lee et al. demonstrated the presence of vascular-derived stem cells in a ruptured human ACL, highlighting the healing-promoting capabilities of CD34+ cells derived from the ACL [17].

Myofibroblasts can have diverse origins despite their development following a consistent sequence [18]. Following injury, fibroblasts exposed to inflammatory mediators (cytokines, mechanical microenvironment) are known to embrace the protomyofibroblast phenotype, subsequently transforming into typical myofibroblasts characterized by the de novo expression of α-SMA, the predominant actin isoform in vascular smooth muscle cells [19]. Myofibroblasts’ contractile activity is significantly enhanced by the expression of α-SMA [18]. In a recent study, Menetrey et al. [20] reported α-SMA-positive cells in the MCL on the third day after the injury, migrating towards the center of the lesion. At the same time, the healing ACL showed only low levels of α-SMA expression. The diminished density of myofibroblasts in the healing ACL may correspond to the marginally limited number of precursor cells at the injury site [21]. Kanaya et al. demonstrated that mesenchymal stromal cells injected into partially torn rat ACL resulted in an overall improved healing process [21].

The novel EL theory has brought to light a notable disparity in the healing capacities between the proximal and distal ends of the ACL. This naturally prompts the question: Are there morphological and/or molecular expression differences that could account for the observed variation in healing potential between the proximal and the distal part of the ligament? To address this inquiry and to enhance the EL theory, we comprehensively examined the human ACL’s proximal and distal segments. We assessed the quantity and distribution of cells in the EL and performed immunohistochemical analysis to determine the expression of CD34 and α-SMA. The identified expression differences between the two ACL segments were then correlated with the EL theory.

## 2. Materials and Methods

### 2.1. Tissue Preparation

For histology and immunohistochemistry, we obtained samples from the proximal and distal ends of the ACLs from 12 fresh knee joints of five fresh male and seven female European cadavers in the Department of Anatomy, Histology, and Embryology at the Medical University of Sofia. The mean age of the cadavers at death was 55 years (min 49 years; max 62 years). No clinical data indicated previous trauma, knee osteoarthritis, scars around the knee, or surgical history. The authors state that every effort was made to follow all local and international ethical guidelines and laws that pertain to the use of human cadaveric donors in anatomical research [22].

Samples were acquired from the proximal and distal parts of the ACL and fixed according to the established protocol for routine histological and immunohistochemical examination, described in detail in our previous study [12].

### 2.2. Light Microscopy

For light microscopy, 5 μm thick sections were cut on a microtome (Leica, Wetzlar, Germany). The paraffin sections were mounted on microscope slides and routinely stained with Mallory’s trichrome stain and hematoxylin and eosin according to the standard methods described in our previous studies [11,12].

### 2.3. Immunohistochemistry

Multiple specimens were fixed in 10% buffered formalin, embedded in paraffin, and cut to 4 μm thickness. Our previous study presented the well-established routine protocol for immunohistochemical work in detail [12].

The following antibodies were used for immunohistochemistry: monoclonal mouse anti-human α-SMA (M0851, DAKO, Agilent Technologies, Inc., Santa Clara, CA, USA) and monoclonal mouse anti-human CD34 (M7165, DAKO, Agilent), both diluted 1:100. The detection system was EnVision™ FLEX+, Mouse, High pH (Link) (K8002, DAKO Cytomation, Agilent, Glostrup, Denmark). The manufacturer’s protocols were followed. Eighteen sections were used as controls.

Photomicrographs of representative immunohistochemical staining fields were obtained using an Olympus CX 21 microscope fitted with an Olympus C5050Z digital camera (Olympus Optical Co., Ltd., Tokyo, Japan).

### 2.4. Semiquantitative Analysis

For semiquantitative analysis of CD34 and α-SMA expression, we used ImageJ 1.53f software, which was free to download from the website of the National Institute of Health (NIH) (http://imagej.nih.gov/ij/, accessed on 30 November 2020) [13]. The intensity of staining was assessed through the IHC Profiler plugin, another free download from the Sourceforge website (https://sourceforge.net/projects/ihcprofiler/, accessed on 30 November 2020), according to a well-established protocol [14]. The IHC profiler assigned a score to each visual field on a four-tier system: high positive (3+), positive (2+), low positive (1+), and negative (0). Five slides from each ligament were studied. We analyzed at least ten random visual fields on each slide. The final score was the average of the scores of all visual fields as calculated by the IHC Profiler.

### 2.5. Quantification of Cell Numbers

To measure cell density, we subjected the images described above to expert-trained supervised machine learning-based pixel classification in ilastik software, version 1.3.3post3 [23]. To distinguish pixels of tissue area, background, and nuclei, we trained the classifier to provide results comparable to those a histology expert gave across various example images. The segmentation masks of nuclei were then subjected to particle analysis after setting a size and shape limit and filtering out noise and debris. The resulting counts were normalized to tissue area masks of the same image. The results were expressed as cell counts per square millimeter.

The image count data were analyzed using the statistical programming language R v4.2.2 [24] and the integrated development environment R Studio v2023.03.0+386 [25]. Multiple t tests with a Holm–Sidak correction, assuming similar dispersions, were used for group comparisons. Values of *p* < 0.05 were considered statistically significant. Graphic data representations were generated by the package ggplot2 v3.4.0 for data visualization [26].

## 3. Results

### 3.1. Light Microscopy

The EL is morphologically similar in the proximal (Figure 1) and distal (Figure 2) parts of the ACL, comprising various types of connective tissue cells, including active fibroblasts, nonactive fibroblasts (fibrocytes), and adipocytes. There were also extracellular collagen fibers with uniformly small diameters, singly or in groups, in the EL tissue of both ligaments; most of the neurovascular bundles of the EL-ligament complex were predominantly located in the EL. There were more fibroblasts in the EL of the proximal and distal parts of the ACL than in the EL of the midsubstance ACL, as quantified in our previous study [12]. There were also more cells in the EL of the two ends of the ACL than in the overall ligament substance. The EL morphology was strikingly similar to that of the synovium.

### 3.2. Expression of CD34 and α-SMA in Proximal and Distal ACL EL

In the EL of the proximal and distal parts of the ACL, immunostaining for CD34 was strongest in the endothelial layers of the blood vessels and was uniformly expressed in the EL tissue (Figure 3A,B and Figure 4A,B). Immunostaining for α-SMA was strongest in the smooth muscle cells of the tunica media of blood vessels, and there was no reaction in the superficial layer of the EL of the ACL (Figure 3C,D and Figure 4C,D), in contrast to the midsubstance, as reported in our previous study [12].
Figure 1Normal morphology of the EL of the proximal ACL in the human knee. EL—epiligament; L—ligament tissue; small arrowhead—blood vessels in the EL; large arrowhead—EL extending towards the endoligament. (**A**,**B**). Hematoxylin and eosin stain; (**C**,**D**). Mallory’s trichrome stain. Scale bar, 200 μm.
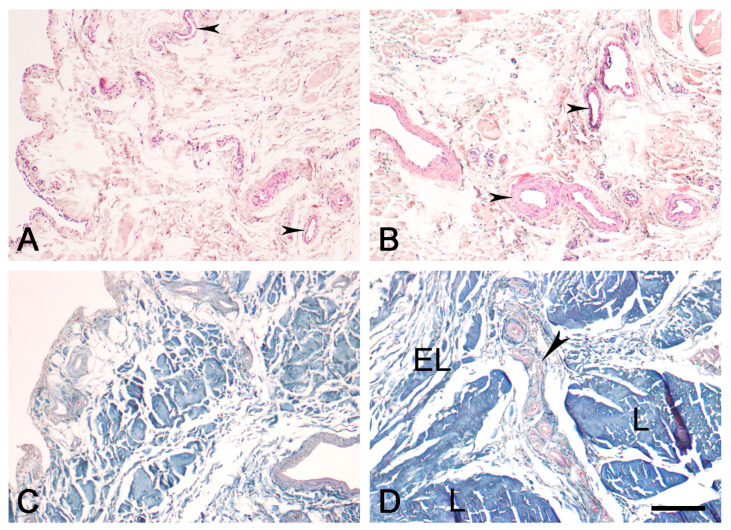

Figure 2Normal morphology of the EL of the distal ACL in the human knee. EL—epiligament; L—ligament tissue; arrows—blood vessels in the EL; asterisk—EL extending towards the endoligament. (**A**,**B**). Hematoxylin and eosin stain; (**C**,**D**). Mallory’s trichrome stain. Scale bar, 200 μm.
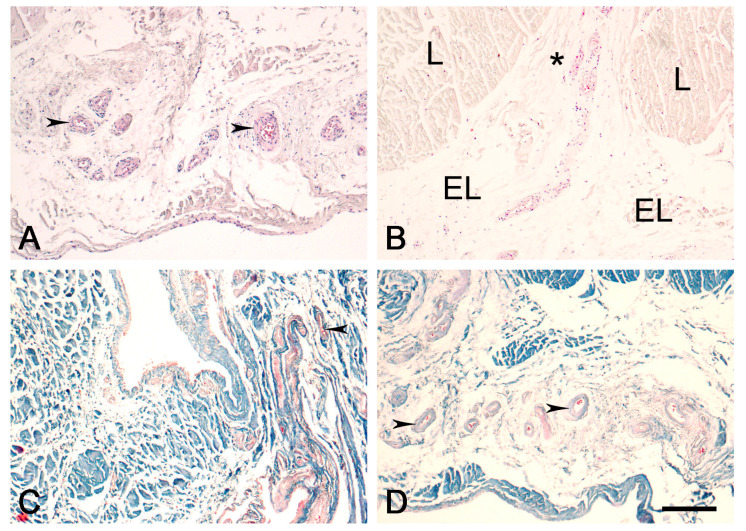


The intensities of the immunohistochemical reactions for CD34 and α-SMA in the proximal and distal parts of the ACL were calculated using the IHC Profiler Plugin for ImageJ (Table 1). Two parameters were considered: the number of images with a specific overall score (Figure 5) and the distribution of visual fields with specific scores across images of the same type of EL stained with the same marker (Table 1).

The immunohistochemical staining results obtained by the IHC Profiler were analyzed by obtaining two correlated statistics: the fraction of images with a specific overall score and average percentages of image areas with the respective score across all images from the experimental group.

Across the images analyzed, DAB staining for CD34 and α-SMA showed more positive fields overall (scores 1, 2 or 3, excluding 0 scores) in the proximal than the distal third of the EL of the ACL. However, with CD34 DAB staining, there were more positive (2) and high positive (3) fields in the distal than the proximal third (Table 1). This was confirmed by the distribution of images with overall IHC scores (Figure 5). There was a small fraction of positively scored (2) images in the distal third of the EL of the ACL, while all analyzed images of the proximal third had low positive (1) overall scores for CD34 (Figure 5). In α-SMA DAB-stained images, there was an overall trend toward higher levels of this marker in the proximal third of the EL of the ACL than in the distal third.

The cell density results were obtained by automated image analysis based on nuclei counts, normalized by the total tissue area fraction in the image (Figure 6). The cell counts in the EL of the ACL tended to be higher and more varied than those in the respective ligament part (distal parts: *p* = 0.00097; proximal parts: *p* = 0.27). The median cell counts in the proximal and distal thirds of the EL were higher than the median value in the medial third of the EL ACL, which is statistically significant when comparing the distal parts of the EL and ligament (*p* = 0.00096) (Figure 6) [12]. The cell counts were slightly higher in the distal than the proximal EL ACL, but the difference was not statistically significant (*p* = 0.425). All the above-mentioned *p* values were adjusted for multiple testing by the Holm–Sidak method.

## 4. Discussion

To elaborate on the literature data concerning the EL theory about failure of ACL healing after injury to the middle part, and the results after suture and reinsertion of the ends of the ACL, we aimed to (1) analyze the number of cells in the proximal and distal parts of the ACL morphologically and quantitatively; (2) evaluate differences in immunohistochemical expression of CD34 and α-SMA, which have definite roles in ligament healing; and (3) determine whether the differences revealed by (1) and (2) could explain the better regenerative capacity of the proximal and distal parts of the ACL.

In the current article, our first aim was to determine whether morphological descriptions and quantification of cell numbers in the proximal and distal parts of the ACL EL could develop the EL theory further and explain why the healing capacity of those parts differs from that of the midportion ACL. Our answer is “yes” because (1) the EL is the main depot of cells and neurovascular bundles in the EL-ligament complex in the parts of the ACL investigated, and (2) there are more cells in the proximal and distal ACL EL than in the midsubstance, as described in our previous study [11]. Therefore, it can be inferred that this connective tissue layer is the primary donor of cells and blood vessels during healing and could explain why the regenerative capacity of the regions investigated is better than that of the midportion of the ACL.

Our second aim was to evaluate the differences in location and immunohistochemical expression of CD34 and α-SMA, which have definite roles in ligament healing. However, what is our current understanding of the sophisticated roles these molecules play in the healing of ligaments?

The walls of blood vessels are recognized as a rich source of stem/progenitor cells that express CD34 surface markers [15,16]. These circulating human CD34 cells play a crucial role in angiogenesis within an environment induced by MCL injury, making a noteworthy contribution to the morphological healing of ligaments. Within ACL tissue, CD34-expressing vascular cells exhibit the potential for multilineage differentiation and the ability to migrate to the site of ACL ruptures, contributing to ligament healing [27]. In a dog model of ACL reconstruction, Matsumoto et al. [28] examined the maturation of bone–tendon integration. They observed endochondral ossification-like integration with enhanced angiogenesis in tissue grafts treated with CD34+ cells. The researchers also confirmed the recruitment of CD34+ cells to the rupture site, noting a significantly higher presence of these cells in that area compared to the midsubstance region. Using CD34+ endothelial progenitor cells from ACL remnants in human patients undergoing surgery, Mifune et al. [29] improved bone–tendon healing in a rat model of ACL rupture. Subsequently, Mifune et al. [30] demonstrated that ACL-derived CD34+ cells contributed to bone–tendon healing through angiogenesis and enhanced osteogenesis. Moreover, the authors reported a higher abundance of vascular-derived CD34+ stem cells in the remnants of ruptured ACL tissue compared to the uninjured midsubstance of the ACL. Nakano et al. [31] conducted a study demonstrating that cells derived from the ACL of a younger group improved bone–tendon healing in an immunodeficient rat model of ACL reconstruction. Kirizuki et al. explored the potential for ACL healing by analyzing morphological patterns. They observed a significantly higher presence of CD34+ cells in the non-reattachment group compared to the reattachment group [32].

Myofibroblasts, characterized by the presence of α-SMA, have been identified in intact and injured human ACLs [33,34]. In the midsubstance of the intact human ACL, Murray and Spector [34] demonstrated the existence of myofibroblast-like cells. Murray et al. [33] later suggested that rather than individual myofibroblasts dispersed throughout the tissue, a continuous layer of α-SMA-containing cells surrounding the tissue could be responsible for contraction, leading to the retraction of ACL remnants. Additionally, the development of a synovial layer consisting of cells that possess a contractile actin isoform over the EL tissue may play a role in the retraction of the remnants, hindering the formation of reparative bridging tissue [33].

Nevertheless, what does our study unveil regarding the expression of these molecules, and how can the EL theory elucidate it?

Firstly, CD34 was similarly expressed in the proximal and distal thirds of the EL of the ACL, although it seemed to be slightly more pronounced in the distal third. These results are in line with the findings from our previous study of the EL of the midportion ACL: The reaction for CD34 was positive in the endothelia and uniformly expressed in the EL tissue. We, therefore, considered that the EL tissue is the primary source of CD34+ cells in the midsubstance of the ACL; these are accepted as the stem/progenitor cells needed for subsequent ligament healing [15,16]. The expression of CD34+ cells in the proximal and distal ACL EL and the higher number of cells suggests that the EL tissue will ensure better healing at the ends of the ACL. We accept the statement by Kirizuki et al. that remnant-preserving ACL surgery to the proximal and distal parts of the ACL, based on hypercellular EL tissue, could be the leading donor of CD34+ cells and therefore ensure better incorporation and healing of the graft [32].

Secondly, α-SMA showed an overall trend toward higher expression at both ends of the ACL EL than in the midsubstance of the ACL EL [12], which showed more negatively (0) scored fields. α-SMA was detected predominantly in the smooth muscle cells of the blood vessels. In contrast to the midsubstance of ACL EL [11], there was a lack of reaction for α-SMA in the superficial layer of the proximal and distal ACL EL. This layer comprises cells that express a contractile actin isoform over the EL, partly explaining the retraction of the midsubstance after injury, which disfavors reparative bridging tissue [33]. In the proximal and distal ACL EL, despite higher overall antibody expression, this layer does not express α-SMA, so the torn ends have lower retraction capacity. This, together with the higher cell count in the proximal and distal parts than the middle part reported by Georgiev et al., augments the EL theory and explains the better healing capacity of the proximal and distal parts of the ACL together with lower retraction of the ends [11].

Of course, although the data presented about the proximal and distal third ACL EL based on the EL theory augments the literature data, many explanations exist for ACL healing failure. As we stated previously, ACL healing failure is multifactorial, and no single theory can fully explain an inadequate healing process [12,14].

This study has limitations: (1) The age of the cadavers could compromise the results owing to age-related alterations [35,36,37]. Therefore, we used fresh cadavers with a mean age at death of 55 years with no previous history of osteoarthritis or trauma. (2) Visual quantification of immunohistochemical images is susceptible to significant inter- and intra-observer variations. To eliminate this, we used the IHC Profiler plugin for ImageJ software. (3) Only healthy (normal) ACL EL was investigated. Future studies need to be performed on injured ACL EL.

## 5. Conclusions

Our results demonstrate, for the first time, the morphology of the ACL EL in the proximal and distal parts and analyze their differences from the ACL midportion in terms of cell numbers and the expression of CD34 and α-SMA. These molecules are involved in ligament healing and are expressed mainly in the blood vessel walls. It should also be noted that the EL is the major source of blood vessels to the ligament-EL complex and that the expression of the aforementioned molecules, together with the number of cells in the proximal and distal parts of the ACL EL, is higher than in the midsubstance. These results augment the EL theory and could, therefore, explain: (1) the better susceptibility to repair of proximal and distal ACL lesions; (2) remnant-preserving ACL reconstruction should be performed because the EL tissue is the leading donor of CD34+ cells and will, therefore, ensure better healing. Future investigations to enhance our understanding of ACL healing should not neglect the EL theory.

## Figures and Tables

**Figure 3 biomedicines-12-00156-f003:**
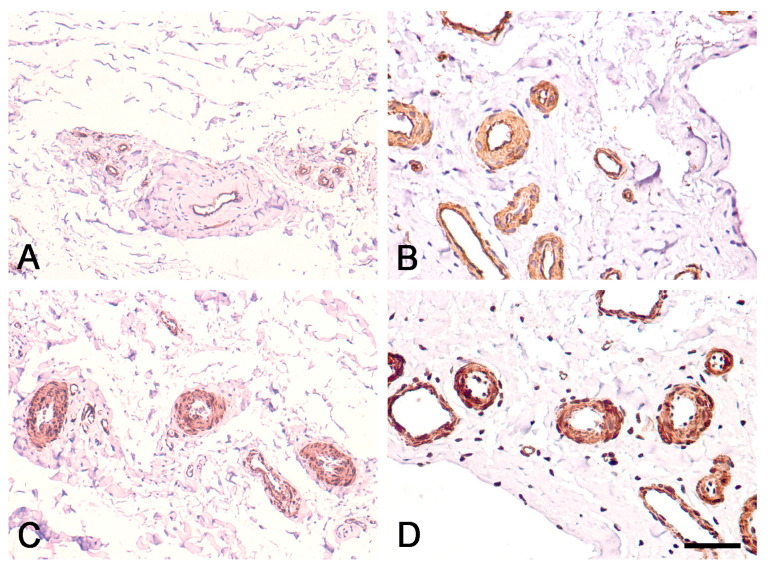
Immunohistochemical staining for CD34 and α-SMA in the EL of the proximal ACL in the human knee. (**A**,**B**): Immunohistochemical staining for CD34. (**A**) Scale bar, 100 μm; (**B**) Scale bar, 50 μm; (**C**,**D**). Immunohistochemical staining for α-SMA. (**C**) Scale bar, 100 μm; (**D**) Scale bar, 50 μm.

**Figure 4 biomedicines-12-00156-f004:**
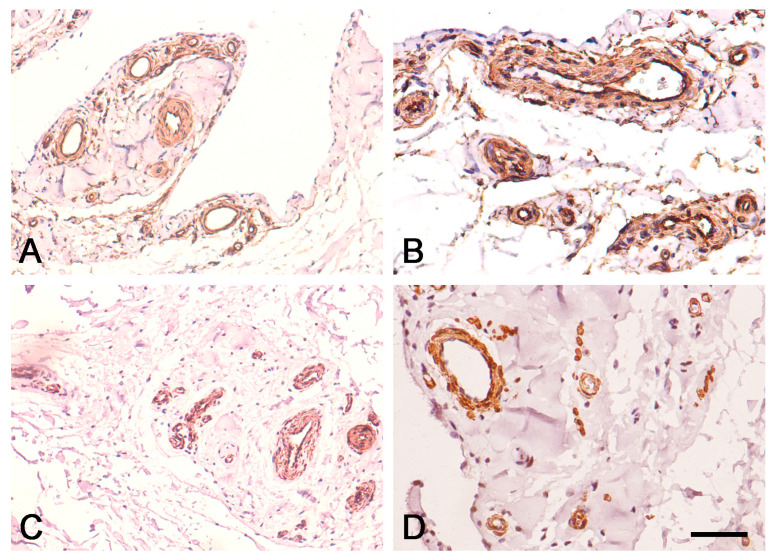
Immunohistochemical staining for CD34 and α-SMA in the EL of the distal ACL in the human knee. (**A**,**B**) Immunohistochemical staining for CD34. (**A**) Scale bar, 100 μm; (**B**) Scale bar, 50 μm; (**C**,**D**). Immunohistochemical staining for α-SMA. (**C**) Scale bar, 100 μm; (**D**) Scale bar, 50 μm.

**Figure 5 biomedicines-12-00156-f005:**
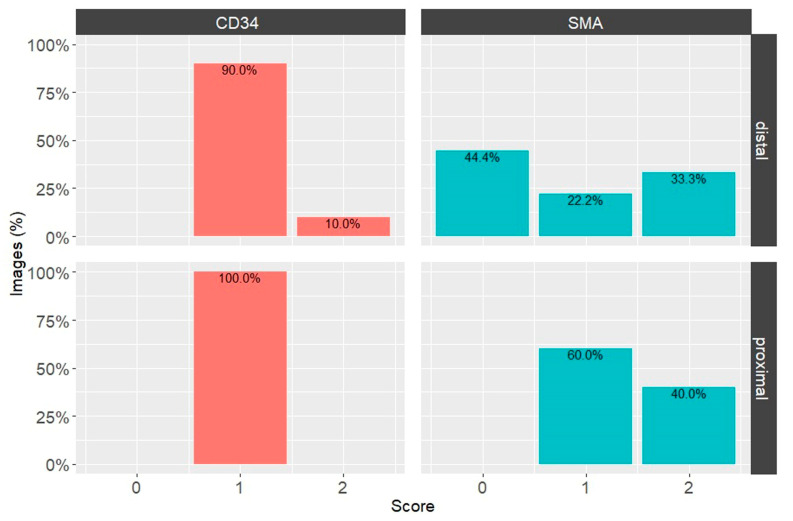
Percentages of overall image grades, according to ICH Profiler-generated scores for CD34 and α-SMA in the EL of the ACL in the human knee. Bars are color-coded according to the immunohistochemical staining: CD34—orange, α-SMA—green. Top row—distal EL images; bottom row—proximal EL images.

**Figure 6 biomedicines-12-00156-f006:**
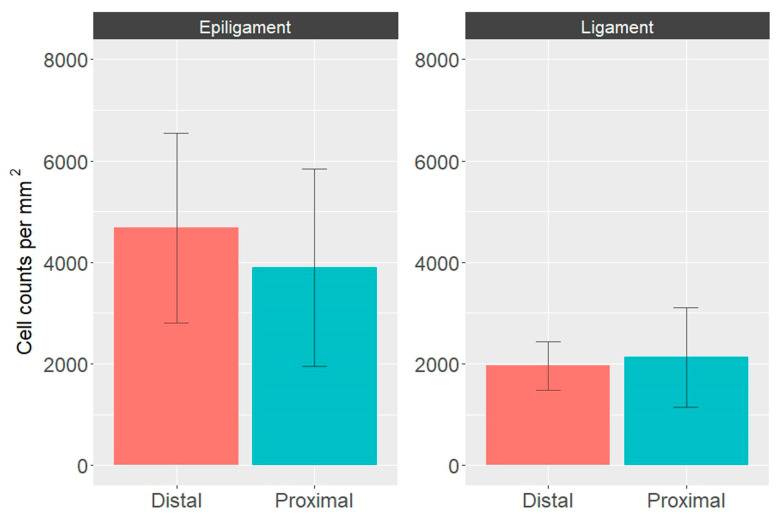
Quantitative cell density data in histological slice images of EL and ligament tissue of ACL. Bar plots are color-coded according to the location of the sample; orange for the distal third, blue for the proximal third. Data are presented as mean and SD.

**Table 1 biomedicines-12-00156-t001:** Semi-quantitative analysis of the immunohistochemical expression of CD34 and α-SMA in the EL of the proximal third and distal third of the anterior cruciate ligament. The percentage for each score represents the percentage of visual fields to which the IHC Profiler assigned this score.

Proximal Third	Distal Third	IHC Marker
High Positive (3+) (3.8%)	High Positive (3+) (4.0%)	CD34
Positive (2+) (9.4%)	Positive (2+) (9.7%)	
Low Positive (1+) (78.9%)	Low Positive (1+) (55.0%)	
High Positive (3+) (7.4%)	High Positive (3+) (3.7%)	α-SMA
Positive (2+) (22.5%)	Positive (2+) (27.1%)	
Low Positive (1+) (69.3%)	Low Positive (1+) (20.2%)	
Negative (0) (0.8%)	Negative (0) (49.0%)	

## Data Availability

Please contact the authors for data requests.

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
