# Peer review of "Do the Differences in the Epiligament of the Proximal and Distal Parts of the Anterior Cruciate Ligament Explain Their Different Healing Capacities? Quantitative and Immunohistochemical Analysis of CD34 and α-SMA Expression in Relation to the Epiligament Theory"

_biomedicines, 2024, doi:10.3390/biomedicines12010156_

Round 1
Reviewer 1 Report
Comments and Suggestions for Authors
Interesting article on a little covered topic.
This is a Quantitative and immunohistochemical analysis of CD 34 and α-SMA expression in relation to the epiligament theory
The primary aim was to Quantitative and immunohistochemical analysis of CD 34 and α-SMA expression in relation to the epiligament theory
There are few related articles.
ABSTRACT:
Please write in detail whether the differences in cell density and cell counts were statistically significant. Introduction: Please do a larger literature review. Please cite and discuss more articles related to the topic. Please describe why CD 34 and α-SMA expression affects the regeneration and properties of the ACL (please support this with articles). Please put forward a research hypothesis. Please write clearly the purpose of the work. Materials and methods clearly and extensively presented. Results: Well presented. Clear, legible and nice tables and figures. Discussions: Please do a larger literature review. Please cite and discuss more articles related to the topic. Please describe why CD 34 and α-SMA expression affects the regeneration and properties of the ACL (please support this with articles). Please expand the limitations. Please write how the results of your research may be useful in orthopedic practice. Conclusions: Please add one practical conclusion for orthopedists.
Author Response
January 03, 2024
Dear Editor,
Herein, we resubmit the revised version of our article entitled “Do the differences in the epiligament of the proximal and distal parts of the anterior cruciate ligament explain their different healing capacities? Quantitative and immunohistochemical analysis of CD 34 and α-SMA expression in relation to the epiligament theory” (biomedicines-2804157) by Georgiev et al.
According to the Reviewers’ comments, we have rewritten the manuscript and we would like to resubmit it for publication.
Details of the changes made are included below.
We added further details about the results of our paper, and highlighted their statistical significance.
Introduction: we improved the literature review by citing more papers, as requested by the reviewers. Moreover, we provided indepth information about the role of CD 34 and α-SMA in the healing process of ACL. We also formulated a clear hypothesis describing the purpose of our work.
Discussion: we further discussed the included additional papers in the literature review and thoroughly discussed why CD 34 and α-SMA expression affects the regeneration and properties of the ACL, supporting our claims with proper articles. Furthermore, we provided practical uses of our results for the orthopedics practice and further broadened the stated limitations of our work.
Conclusion: we improved the conclusion so it is practical for orthopedists.
The changes in the text are marked in red.
Response to reviewers' comments: first we want to express our utmost gratitude to the reviewers for taking their time and providing us with constructive criticism and valuable suggestions on how to improve our article.
Reviewer #1:
ABSTRACT:
Please write in detail whether the differences in cell density and cell counts were statistically significant.
we accept the suggestion and have provided the necessary information.
Introduction: Please do a larger literature review. Please cite and discuss more articles related to the topic. Please describe why CD 34 and α-SMA expression affects the regeneration and properties of the ACL (please support this with articles). Please put forward a research hypothesis. Please write clearly the purpose of the work.
We accept all of the mentioned suggestions and have fulfilled them best to our ability.
Materials and methods clearly and extensively presented.
Results: Well presented. Clear, legible and nice tables and figures.
Discussions: Please do a larger literature review. Please cite and discuss more articles related to the topic. Please describe why CD 34 and α-SMA expression affects the regeneration and properties of the ACL (please support this with articles).
Please expand the limitations.
Please write how the results of your research may be useful in orthopedic practice.
we accept all of the above stated corrections and have incorporated all of the suggested correction in our revised manuscript.
Conclusions: Please add one practical conclusion for orthopedists.
we accept the suggestion and have improved our conclusion.
Reviewer #2:
This is a very interesting basic study on ACL healing, showing that the ACL’s epiligament resembles the synovium. The study may have an impact on the functional capacity of athletic populations post-ACL injury & repair.
However, the fact remains that it is a histological and immunohistochemical study on 12 cadaveric knees of subjects that had no previous trauma, osteoarthritis or surgical history of their knee joint. The immunohistochemical conditions may be altered post ACL injury, therefore results of the current study may have to be replicated in such a sample. Also, subjects that are prone to knee injuries may have a different immunohistochemical and histological profile than those not injured.
we agree with this point. The study's reliance on histological and immunohistochemical data from a limited sample of 12 cadaveric knees, all without prior trauma, osteoarthritis, or surgical interventions, is a potential limitation. The immunohistochemical conditions might indeed differ post-ACL injury, suggesting the need for replication in a sample that includes individuals with ACL injuries. Additionally, individuals prone to knee injuries may exhibit distinct immunohistochemical and histological profiles compared to those without injuries. This consideration highlights the importance of exploring a more diverse and injury-prone population in future studies to draw comprehensive conclusions about ACL healing mechanisms.
We added this as limitation of the study.
Nevertheless, the authors are to be commended for their detailed methodology and analysis of the ACL distal and proximal specimens obtained.
A question to the authors is why were there not samples of the epiligament taken from the middle portion of the ACL (lines 69-70) in this study also. However, the authors then state in lines 228-9 that: “We confirmed the results from our previous study: in the EL of the midpart ACL, the reaction for CD34 was positive in the endothelia and uniformly expressed in the EL tissue.” How can this be ‘confirmed’ if not measured in this study?
The sentence was rephrased.
Reviewer 2 Report
Comments and Suggestions for Authors
This is a very interesting basic study on ACL healing, showing that the ACL’s epiligament resembles the synovium. The study may have an impact on the functional capacity of athletic populations post-ACL injury & repair.
However, the fact remains that it is a histological and immunohistochemical study on 12 cadaveric knees of subjects that had no previous trauma, osteoarthritis or surgical history of their knee joint. The immunohistochemical conditions may be altered post ACL injury, therefore results of the current study may have to be replicated in such a sample. Also, subjects that are prone to knee injuries may have a different immunohistochemical and histological profile than those not injured.
Nevertheless, the authors are to be commended for their detailed methodology and analysis of the ACL distal and proximal specimens obtained.
A question to the authors is why were there not samples of the epiligament taken from the middle portion of the ACL (lines 69-70) in this study also. However, the authors then state in lines 228-9 that: “We confirmed the results from our previous study: in the EL of the midpart ACL, the reaction for CD34 was positive in the endothelia and uniformly expressed in the EL tissue.” How can this be ‘confirmed’ if not measured in this study?
Line 51: It seems that a word or two are missing after the word ‘cartilage’.
Comments on the Quality of English LanguageVery good.
Author Response

(The authors gave the same response as above.)

Round 2
Reviewer 1 Report
Comments and Suggestions for Authors The authors made the suggested changes. Manuscript for publication.